# Hierarchical Selective Classification

**Shani Goren\***
Technion
shani.goren@gmail.com

**Ido Galil\***
Technion, NVIDIA
idogalil.ig@gmail.com, igalil@nvidia.com

**Ran El-Yaniv**
Technion, NVIDIA
rani@cs.technion.ac.il, relyaniv@nvidia.com

## Abstract

Deploying deep neural networks for risk-sensitive tasks necessitates an uncertainty estimation mechanism. This paper introduces *hierarchical selective classification* (HSC), extending selective classification to a hierarchical setting. Our approach leverages the inherent structure of class relationships, enabling models to reduce the specificity of their predictions when faced with uncertainty. In this paper, we first formalize hierarchical risk and coverage, and introduce hierarchical risk-coverage curves. Next, we develop algorithms for performing HSC (which we refer to as "inference rules"), and propose an efficient algorithm that guarantees a target accuracy constraint with high probability. We also introduce Calibration-Coverage curves, which analyze the effect of hierarchical selective classification on calibration. Our extensive empirical studies on over a thousand ImageNet classifiers reveal that training regimes such as CLIP, pretraining on ImageNet21k, and knowledge distillation boost hierarchical selective performance. Lastly, we show that HSC improves both selective performance and confidence calibration. Code is available at `https://github.com/shanigoren/Hierarchical-Selective-Classification`.

## 1 Introduction

Deep neural networks (DNNs) have achieved incredible success across various domains, including computer vision and natural language processing. To ensure the reliability of models intended for real-world applications we must incorporate an uncertainty estimation mechanism, such as *selective classification* [19], which allows a model to abstain from classifying samples when it is uncertain about their predictions. Standard selective classification, however, has an inherent shortcoming: for any given sample, it is limited to either making a prediction or rejecting it, thereby ignoring potentially useful information about the sample, in case of rejection.

To illustrate the potential consequences of this limitation, consider a model trained for classifying different types of brain tumors from MRI scans, including both benign and malignant tumors. If a malignant tumor is identified with high confidence, immediate action is needed. For a particular hypothetical scan, suppose the model struggles to distinguish between 3 subtypes of malignant tumors, assigning a confidence score of 0.33 to each of them (assuming those estimates are well-calibrated and sum to 1). In the traditional selective classification framework, if the confidence threshold is higher than 0.33, the model will reject the sample, failing to provide valuable information to alert healthcare professionals about the patient's condition, even though the model has enough information to conclude that the tumor is malignant with high certainty. This could potentially result in delayed diagnosis and treatment, posing a significant risk to the patient's life. However, a hierarchically-aware selective model with an identical confidence threshold would classify the tumor as malignant with

---

*The first two authors have equal contribution.

38th Conference on Neural Information Processing Systems (NeurIPS 2024).

99% confidence. Although this prediction is less specific, it remains highly valuable as it can promptly notify healthcare professionals, leading to early diagnosis and life-saving treatment for the patient. This motivates us to propose *hierarchical selective classification* (HSC), an extension of selective classification to a setting where the classes are organized in a hierarchical structure. Such hierarchies are typically represented by a tree-like structure, where each node represents a class, and the edges reflect a semantic relationship between the classes, most commonly an 'is-a' relationship. Datasets with an existing hierarchy are fairly common, as there are many well-established predefined hierarchies available, such as WordNet [33] and taxonomic data for plants and animals. Consequently, popular datasets like ImageNet [11], which is widely used in computer vision tasks, and iNaturalist [24], have been built upon these hierarchical structures, providing ready-to-use trees. The ImageNet dataset, for example, is organized according to the WordNet hierarchy. A visualization of a small portion of the ImageNet hierarchy is shown in Figure 1. In cases where a hierarchical tree structure is not provided with the dataset, it can be automatically generated using methods such as leveraging LLMs [7, 17, 28, 57].

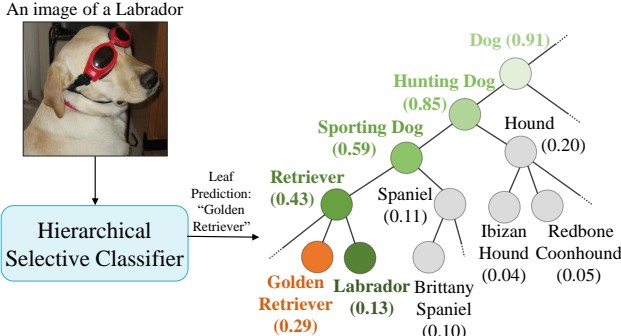

Figure 1: A detailed example of HSC for the output of ViT-L/16-384 on a specific sample. The base classifier outputs leaf softmax scores, with internal node scores being the sum of their descendant leaves' scores, displayed in parentheses next to each node. The base classifier incorrectly classifies the image as a 'Golden Retriever' with low confidence. A selective classifier can either make the same incorrect leaf prediction if the confidence threshold is below 0.29, or reject the sample. A hierarchical selective classifier with the *Climbing* inference rule (see Section 3) climbs the path from the predicted leaf to the root until the confidence threshold $\theta$ is met. Setting $\theta$ above 0.29 yields a hierarchically correct prediction, with smaller $\theta$ values increasing the coverage. An Algorithm for determining the optimal threshold is introduced in Section 4.

The key contributions of this paper are as follows:
(1) **We extend selective classification to a hierarchical setting.** We define *hierarchical selective risk* and *hierarchical coverage*, leading us to introduce *hierarchical risk-coverage curves*.
(2) **We introduce *hierarchical selective inference rules***, i.e., algorithms used to hierarchically reduce the information in predictions based on their uncertainty estimates, improving hierarchical selective performance compared to existing baselines. We also identify and define useful properties of inference rules.
(3) **We propose a novel algorithm to find the optimal confidence threshold** compatible with any base classifier without requiring any fine-tuning, that achieves a user-defined target accuracy with high probability, which can also be set by the user, greatly improving over the existing baseline.
(4) **We conduct a comprehensive empirical study evaluating HSC on more than 1,000 ImageNet classifiers**. We present numerous previously unknown observations, most notably that training approaches such as pretraining on larger datasets, contrastive language-image pretraining (CLIP) [35], and knowledge distillation significantly boost selective hierarchical performance. Furthermore, we define *hierarchical calibration* and discover that HSC consistently improves confidence calibration.

## 2 Problem Setup

**Selective Classification:** Let $\mathcal{X}$ be the input space, and $\mathcal{Y}$ be the label space. Samples are drawn from an unknown joint distribution $P(\mathcal{X} \times \mathcal{Y})$ over $\mathcal{X} \times \mathcal{Y}$. A classifier $f$ is a prediction function $f : \mathcal{X} \to \mathcal{Y}$, and $\hat{y} = f(x)$ is the model's prediction for $x$. The *true risk* of a classifier $f$ w.r.t. $P$

is defined as: $R(f|P) = \mathbb{E}_P[\ell(f(X), Y)]$, where $\ell : \mathcal{Y} \times \mathcal{Y} \to \mathbb{R}^+$ is a given loss function, for instance, the 0/1 loss. Given a set of labeled samples $S_m = \{(x_i, y_i)\}_{i=1}^m$, the *empirical risk* of $f$ is: $\hat{R}(f|S_m) = \frac{1}{m} \sum_{i=1}^m \ell(f(x_i), y_i)$. Following the notation of [21], we use a *confidence score* function $\kappa(x, \hat{y}|f)$ to quantify prediction confidence. We require $\kappa$ to induce a partial order over instances in $\mathcal{X}$. In this work, we focus on the most common and well-known $\kappa$ function, *softmax response* [10, 40]. For a classifier $f$ with softmax as its last layer: $\kappa(x, \hat{y}|f) = f_{\hat{y}}(x)$. Softmax response has been shown to be a reliable confidence score in the context of selective classification [18, 19] as well as in hierarchical classification [46], consistently achieving solid performance.

A *selective model* [9, 13] is a pair $(f, g)$ where $f$ is a classifier and $g : \mathcal{X} \to \{0, 1\}$ is a *selection function*, which serves as a binary selector for $f$. The selective model abstains from predicting instance $x$ if and only if $g(x) = 0$. The selection function $g$ can be defined by a confidence threshold $\theta$: $g_\theta(x|\kappa, f) = \mathbb{1}[\kappa(x, \hat{y}|f) > \theta]$. The performance of a selective model is measured using selective risk and coverage. *Coverage* is defined as the probability mass of the non-rejected instances in $\mathcal{X}$: $\phi(f, g) = \mathbb{E}_P[g(X)]$. The *selective risk* of $(f, g)$ is: $R(f, g) = \frac{\mathbb{E}_P[\ell(f(X), Y)g(X)]}{\phi(f, g)}$.

Risk and coverage can be evaluated over a labeled set $S_m$, with the *empirical coverage* defined as: $\hat{\phi}(f, g|S_m) = \frac{1}{m} \sum_{i=1}^m g(x_i)$, and the *empirical selective risk*: $\hat{R}(f, g|S_m) = \frac{\frac{1}{m} \sum_{i=1}^m \ell(f(x_i), y_i)g(x_i)}{\hat{\phi}(f, g|S_m)}$. The performance profile of a selective classifier can be visualized by a *risk-coverage curve* (RC curve) [13], a curve showing the risk as a function of coverage, measured on a set of samples. The area under the RC curve, namely *AURC*, was defined by [21] for quantifying selective performance via a single scalar. See Figure 2a for an example of an RC Curve.

**Hierarchical Classification:** Following the notations of [12] and [46], a hierarchy $H = (V, E)$ is defined by a tree, with nodes $V$ and edges $E$, the root of the tree is denoted $r \in V$. Each node $v \in V$ represents a semantic class: the leaf nodes $\mathcal{L} \subseteq V$ are mutually exclusive ground-truth classes, and the internal nodes are unions of leaf nodes determined by the hierarchy. The root represents the semantic class containing all other objects. A sample $x$ that belongs to class $y$ also belongs to the ancestors of $y$. Each node has exactly one parent and one unique path to it from the root node. The set of leaf descendants of node $v$ is denoted by $\mathcal{L}(v)$, and the set of ancestors of node $v$, including $v$ itself, is denoted by $\mathcal{A}(v)$. A hierarchical classifier $f : \mathcal{X} \to V$ labels a sample $x \in \mathcal{X}$ as a node $v \in V$, at any level of the hierarchy, as opposed to a flat classifier, that only predicts leaves.

Given the hierarchy, it is correct to label an image as either its ground truth leaf node or any of its ancestors. For instance, a Labrador is also a dog, a canine, a mammal, and an animal. While any of these labels is technically correct for classifying a Labrador, the most specific label is clearly preferable as it contains the most information. Thus, it is crucial to observe that the definition of hierarchical correctness is incomplete without considering the amount of information held by the predictions. The correctness of a hierarchical classifier $f$ on a set of samples $S_m$ is defined by: $\frac{1}{m} \sum_{i=1}^m \mathbb{1}[f(x_i) \in \mathcal{A}(y_i)]$. Note that when $H$ is flat and does not contain internal nodes, the hierarchical accuracy reduces to the accuracy of a standard leaf classifier. In the context of classification tasks, the goal is usually to maximize accuracy. However, a trivial way to achieve 100% hierarchical accuracy would be to simply classify all samples as the root node. For this reason, a mechanism that penalizes the model for predicting less specific nodes must be present as well. In section 3 we define *coverage*, which quantifies prediction specificity. We aim for a trade-off between the accuracy and information gained by the prediction, which can be controlled according to a user's requirements. A review of related work involving hierarchical classification can be found in section 6.

**Hierarchical Selective Classification:** Selective classification offers a binary choice: either to predict or completely reject a sample. We propose hierarchical selective classification (HSC), a hierarchical extension of selective classification, which allows the model to retreat to less specific nodes in the hierarchy in case of uncertainty. Instead of rejecting the whole prediction, the model can now partially reject a sample, and the degree of the rejection is determined by the model's confidence. For example, if the model is uncertain about the specific dog breed of an image of a Labrador but is confident enough to determine that it is a dog, the safest choice by a classic selective framework would be to reject it. In our setting, however, the model can still provide useful information that the object in the image is a dog (see Figure 1).

A *hierarchical selective classifier* $f^H \triangleq (f, g_\theta^H)$ consists of $f$, a base classifier, and $g_\theta^H : \mathcal{X} \to V$, a hierarchical selection function with a confidence threshold $\theta$. $g_\theta^H$ determines the degree of partial rejection by selecting a node in the hierarchy with confidence higher than $\theta$. Defining $g_\theta^H$ now becomes non-trivial. For instance, $g_\theta^H$ can traverse the hierarchy tree, directly choose a single node,

or follow any other algorithm, as long as the predicted node has sufficient confidence. For this reason, we refer to $g_\theta^H$ as a *hierarchical inference rule*. In section 3 we introduce several inference rules and discuss their properties.

Hierarchical selective classifiers differ from the previously discussed hierarchical classifiers by requiring a hierarchical selection function. In contrast to non-selective hierarchical classifiers, which may produce predictions at internal nodes without a selection function, a hierarchical selective classifier can handle uncertainty by gradually trading off risk and coverage, controlled by $g_\theta^H$. This distinction is crucial because hierarchical selective classifiers provide control over the full trade-off between risk and coverage, which is not always attainable with non-selective hierarchical classifiers.

To our knowledge, controlling the trade-off between accuracy and specificity was only previously explored by [12], who proposed the *Dual Accuracy Reward Trade-off Search* (DARTS) algorithm, which aims to obtain the most specific classifier for a user-specified accuracy constraint. Our approach differs from theirs in that we guarantee control over the full trade-off, while some coverages cannot be achieved by DARTS.

As mentioned in section 2, it is considered correct to label an image as either its ground truth leaf node or any of its ancestors. Thus, we employ a natural extension of the 0/1 loss to define the true risk of a hierarchical classifier $f^H$ with regard to $P$: $R^H(f^H|P) = \mathbb{E}_P[f^H(X) \notin \mathcal{A}(Y)]$, and the empirical risk over a labeled set of samples $S_m$: $\hat{R}^H(f^H|S_m) = \frac{1}{m} \sum_{i=1}^m \mathbb{1}[f^H(x_i) \notin \mathcal{A}(y_i)]$. An additional risk penalizing hierarchical mistake severity is discussed in Appendix A.

To ensure that specific labels are preferred, it is necessary to consider the specificity of predictions. *Hierarchical selective coverage*, which we propose as a hierarchical extension of selective coverage, measures the amount of information present in the model's predictions. A natural quantity for that is entropy, which measures the uncertainty associated with the classes beneath a given node. Assuming a uniform prior on the leaf nodes, the entropy of a node $v \in V$ is $H(v) = -\sum_{v' \in \mathcal{L}(v)} \frac{1}{|\mathcal{L}(v)|} \log(\frac{1}{|\mathcal{L}(v)|}) = \log(|\mathcal{L}(v)|)$. At the root, the entropy reaches its maximum value, $H(r) = \log(|\mathcal{L}|)$, while at the leaves the entropy is minimized, with $H(y) = 0$ for any leaf node $y \in \mathcal{L}$. This allows us to define coverage for a single node $v$, regardless of $g_\theta^H$. We define *hierarchical coverage* (from now on referred to as coverage) as the entropy of $v$ relative to the entropy of the root node: $\phi^H(v) = 1 - \frac{H(v)}{H(r)}$. The root node has zero coverage, as it does not contain any information. The coverage gradually increases until it reaches 1 at the leaves. We can also define true coverage for a hierarchical selective model: $\phi^H(f^H) = \mathbb{E}_P[\phi(f^H(X)]$. The empirical coverage of a classifier over a labeled set of samples $S_m$ is defined as the mean coverage over its predictions: $\hat{\phi}^H(f^H|S_m) = \frac{1}{m} \sum_{i=1}^m \phi(f^H(x_i))$. For a hierarchy comprised of leaves and a root node, hierarchical coverage reduces to selective coverage, where classifying a sample as the root corresponds with rejection.

The hierarchical selective performance of a model can be visualized with a hierarchical RC curve. The area under the hierarchical RC curve, which we term *hAURC*, extends the non-hierarchical AURC, by using the hierarchical extensions of selective risk and coverage. For examples of hierarchical RC curves see Figure 2a and Figure 2b.

**Hierarchical Calibration:** Confidence calibration refers to the task of predicting probability estimates that accurately reflect the true likelihood of correctness. As reducing hierarchical coverage may improve accuracy, it may also improve calibration over the samples the model has not abstained from. To capture this relationship, we introduce the concept of *Calibration-Coverage Curves* (CC curves), which, analogous to RC curves, plot the Expected Calibration Error (ECE) [34] as a function of coverage. see Figure 5 for an example.

## 3 Hierarchical Selective Inference Rules

To define a hierarchical selective model $f^H = (f, g_\theta^H)$ given a base classifier $f$, an explicit definition of $g_\theta^H$ is required. $g_\theta^H$ determines the internal node in the hierarchy $f^H$ retreats to when the leaf prediction of $f$ is uncertain. $g_\theta^H$ requires obtaining confidence for internal nodes in the hierarchy. Since most modern classification models only assign probabilities to leaf nodes, we follow [12] by setting the probability of an internal node to be the sum of its leaf descendant probabilities: $f_v^H(x) = \sum_{y \in \mathcal{L}(v)} f_y(x)$. Unlike other works that assign the sum of leaf descendant probabilities to internal nodes, our algorithm first calibrates leaf probabilities through *temperature scaling*. Since the probabilities of internal nodes heavily rely on the leaf probabilities supplied by $f$, calibration is

---
**Algorithm 1** Climbing Inference Rule
---
**Input:** Classifier $f$, class hierarchy $H = (V, E)$, sample $x_i \in \mathcal{X}$, confidence threshold $\theta$.
**Output:** Predicted node $\hat{v}$.
    Perform temperature scaling [23]
    Obtain $f_v(x_i) \, \forall v \in V$
    $\hat{v} \leftarrow \arg\max_{y \in L} f_y(x_i)$
    **while** $f_{\hat{v}}(x_i) < \theta$ **do**
        $\hat{v} \leftarrow \mathrm{parent}(\hat{v})$
    **end while**
---

beneficial for obtaining more reliable leaf probabilities, and since the internal nodes probabilities are sums of leaf probabilities, this cumulative effect is even more pronounced. Further, [18] found that applying temperature scaling also improves ranking and selective performance, which is beneficial to our objective. In this section, we introduce several inference rules, along with useful theoretical properties. We propose the *Climbing* inference rule (Algorithm 1), which starts at the most likely leaf and climbs the path to the root, until reaching an ancestor with confidence above the threshold. A visualization of the Climbing inference rule is shown in Figure 1.

We compare our proposed inference rules to the following baselines: The first is the *selective inference rule*, which predicts 'root' for every uncertain sample (this approach is identical to standard "hierarchically-ignorant" selective classification). The second hierarchical baseline, proposed by [46], selects the node with the highest coverage among those with confidence above the threshold. We refer to this rule as "*Max-Coverage*" (MC), and its algorithm is detailed in Appendix B.

Certain tasks might require guarantees on inference rules. For example, it can be useful to ensure that an inference rule does not turn a correct prediction into an incorrect one. Therefore, for an inference rule $g_\theta^H$ we define:

1. **Monotonicity in Correctness**: For any $\theta_1 \leq \theta_2$, base classifier $f$ and labeled sample $(x, y)$: $(f, g_{\theta_1}^H)(x) \in \mathcal{A}(y) \Rightarrow (f, g_{\theta_2}^H)(x) \in \mathcal{A}(y)$. Increasing the threshold will never cause a correct prediction to become incorrect.

2. **Monotonicity in Coverage**: For any $\theta_1 \leq \theta_2$, base classifier $f$ and labeled sample $(x, y)$: $\phi^H[(f, g_{\theta_1}^H)(x)] \geq \phi^H[(f, g_{\theta_2}^H)(x)]$. Increasing the threshold will never increase the coverage.

The Climbing inference rule outlined in Algorithm 1 satisfies both monotonicity properties. MC satisfies monotonicity in coverage, but not in correctness. An additional algorithm that does not satisfy any of the properties is discussed in Appendix C.

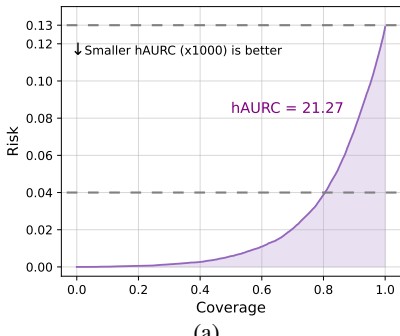 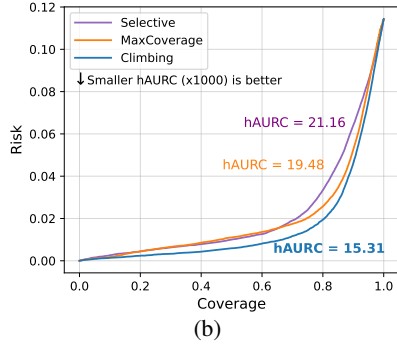

(a)                (b)

Figure 2: (a) hierarchical RC curve of a ViT-L/16-384 model trained on ImageNet1k, evaluated with the 0/1 loss as the risk and softmax response as its confidence function $\kappa$. The purple shaded area represents the area under the RC curve (hAURC). Full coverage occurs when the model accepts all leaf predictions, for which the risk is 0.13. Increasing the confidence threshold leads to the rejection of more samples. For example, when the threshold is 0.77 the the risk is 0.04, with coverage 0.8. (b) hierarchical RC curves of different inference rules with EVA-L/14-196 [14] as the base classifier. When the coverage is 1.0, all inference rules predict leaves. Each inference rule achieves a different trade-off, resulting in distinct curves. This example represents the prevalent case, where the "hierarchically-ignorant" selective inference rule performs the worst and Climbing outperforms MC.

---

**Algorithm 2** Optimal Hierarchical Selective Threshold

---

**Input:** Hierarchical selective classifier $f^H = (f, g^H)$, hierarchy $H = (V, E)$, calibration set $S_n = \{(x_i, y_i)\}_{i=1}^n$, target accuracy $1 - \alpha \in (0, 1)$, confidence level $1 - \delta$.

**Output:** Optimal threshold $\hat{\theta}$, and $\epsilon$ s.t. $P(|C(n, \alpha) - (1 - \alpha)| \leq \epsilon) \geq 1 - \delta$, where $C(n, \alpha) = P(\theta_{n+1} \leq \hat{\theta}|S_n)$

1: Based on $\alpha, n, \delta$, calculate $\epsilon$            ▷ See Appendix E
2: **for** $(x_i, y_i)$ in $S_n$ **do**
3:      Obtain $f_n^H(x_i), \forall v \in V$
4:      $\theta_i \leftarrow (g_\theta^H)^{-1}(x_i|\kappa, f)$        ▷ See Algorithm 5 in Appendix D
5: **end for**
6: Sort $\{\theta_i\}_{i=1}^n$ in ascending order
7: $\hat{\theta} \leftarrow \theta_{\lceil(n+1)(1-\alpha)\rceil}$

---

When comparing hierarchical selective models, we find it useful to measure the performance improvement gained by using hierarchical selective inference. We propose a new metric: *hierarchical gain*, defined as the improvement in hAURC between $(f, g_\theta)$, a "hierarchically-ignorant" selective model, and $(f, g_\theta^{\mathbf{H}})$, the same base classifier with a hierarchical inference rule. This metric might also point to which models have a better hierarchical understanding, as it directly measures the improvement gained by allowing models to leverage hierarchical knowledge. Note that if the selective inference rule is better than the hierarchical inference rule being assessed, the hierarchical gain will be negative. An illustrative individual example of RC curves comparison for one model is shown in Figure 2b.

## 4 Optimal Selective Threshold Algorithm

In Section 3, we defined several inference rules that, given a confidence threshold, return a predicted node in the hierarchy. In this section, we propose an algorithm that efficiently finds the optimal threshold for a user-defined target accuracy and confidence level. The algorithm, outlined in Algorithm 2, receives as input a hierarchical selective classifier $f^H$, an accuracy target $1 - \alpha$, a confidence level $1 - \delta$ (which refers to the interval around $1 - \alpha$, not to be confused with the model's confidence), and a calibration set. It outputs the optimal threshold $\hat{\theta}$ that ensures the classifier's accuracy on an unseen test set falls within a $1 - \delta$ confidence interval around $1 - \alpha$, with a resulting error margin of $\epsilon$. The threshold is calculated once on the calibration set and then used statically during inference. The algorithm does not require any retraining or fine-tuning of the model's weights. For each sample $(x_i, y_i)$ in the calibration set, the algorithm first calculates $\theta_i$, the minimal threshold that would have made the prediction $f^H(x_i)$ hierarchically correct. Then $\hat{\theta}$, the optimal threshold, is calculated in a method inspired by split conformal prediction [48].

**Theorem 1** *Suppose the calibration set $S_n = \{(x_i, y_i)\}_{i=1}^n$ and a given test sample $(x_{n+1}, y_{n+1})$ are exchangeable. For any target accuracy $\alpha \in (0, 1)$ and $\delta$, define $\theta_{n+1}$, $\hat{\theta}$ and $\epsilon$ as in Algorithm 2, and $C(n, \alpha) = P(\theta_{n+1} \leq \hat{\theta}|S_n)$. Then:$P(|C(n, \alpha) - (1 - \alpha)| \leq \epsilon) \geq 1 - \delta$.*
**Proof:** See Appendix E.

**Remark on Theorem 1:** our algorithm can provide an even greater degree of flexibility: the user may choose to set the values of any three parameters out of $\alpha, n, \epsilon, \delta$. With the three parameters fixed, we can compute the remaining parameter. The size of the calibration set is of particular interest. Although increasing $n$ yields more stable results, our guarantees hold for calibration sets of any size. Even for a small calibration set, our algorithm provides users with the flexibility to set the other parameters according to their requirements and constraints. For example: a user with a low budget for a calibration set who may be, perhaps, more interested in controlling $\delta$ can still achieve a reasonable constraint by relaxing $\epsilon$ instead of increasing the calibration set size. See Appendix E for details.

## 5 Experiments

In this section we evaluate the methods introduced in Section 3 and Section 4. The evaluation was performed on 1,115 vision models pretrained on ImageNet1k [11], and 6 models pretrained on iNat-21 [24] (available in timm 0.9.16 [49] and torchvision 0.15.1 [32]). The reported results were

Table 1: Comparison of mean hAURC and hierarchical gain results for the selective, Max-Coverage (MC) and Climbing inference rules applied to 1,115 models trained on ImageNet1k, and 6 models trained on iNat21.

| Inference Rule | ImageNet-1k (1,115 models) | | iNat21 (6 models) | |
|---|---|---|---|---|
| | hAURC ($\times$ 1000) | Hier. Gain (%) | hAURC ($\times$ 1000) | Hier. Gain (%) |
| **Selective** | 42.27$\pm$0.46 | - | 13.35$\pm$0.50 | - |
| **MC** | 39.99$\pm$0.53 | 6.92$\pm$0.26 | 11.96$\pm$0.45 | 10.45$\pm$0.46 |
| **Climbing** | **36.51$\pm$0.47** | **14.94$\pm$0.22** | **10.15$\pm$0.41** | **23.94$\pm$1.24** |

obtained on the corresponding validation sets (ImageNet1k and iNat-21). The complete results and source code necessary for reproducing the experiments are provided in the Supplementary Material.

**Inference Rules:** Table 1 compares the mean results of the Climbing inference rule to both hierarchical and non-hierarchical baselines. The evaluation is based on RC curves generated for 1,115 models pretrained on ImageNet1k and 6 models pretrained on iNat21 with each inference rule applied to their output. These aggregated results show the effect of different hierarchical selective inference rules, independent of the properties or output of a specific model. Compared to the non-hierarchical selective baseline, hierarchical inference has a clear benefit. Allowing models to handle uncertainty by partially rejecting a prediction instead of rejecting it as a whole, proves to be advantageous; the average model is capable of leveraging the hierarchy to predict internal nodes that reduce risk while preserving coverage. Nonetheless, the differences remain stark when comparing the hierarchical inference rules. Climbing, achieving almost 15% hierarchical gain, outperforms MC, with more than double the gain of the latter. These results highlight the fact that the approach taken by inference rules to leverage the hierarchy can have a significant impact. A possible explanation for Climbing's superior performance could stem from the fact that most models are trained to optimize leaf classification. By starting from the most likely leaf, Climbing utilizes the prior knowledge embedded in models for leaf classification, while MC ignores it.

**Optimal Threshold Algorithm:** We compare our algorithm to DARTS [12]. We evaluate the performance of both algorithms on 1,115 vision models trained on ImageNet1k, for a set of target accuracies. For each model and target accuracy the algorithm was run 1000 times, each with a randomly drawn calibration set. We also evaluate both algorithms on 6 models trained on the iNat21 dataset. We compare the algorithms based on two metrics: (1) *target accuracy error*, i.e., the mean distance between the target accuracy and the accuracy measured on the test set; (2) coverage achieved by using the algorithms' output on the test set. The results presented in Table 2, and Table 5 in Appendix G, show that our algorithm consistently achieves substantially lower target accuracy errors, indicating that our algorithm succeeds in nearing the target accuracy more precisely. This property allows the model to provide a better balance between risk and coverage. Our algorithm is more inclined towards this trade-off, as it almost always achieves higher coverage than DARTS. This is particularly noteworthy when the target accuracy is high: while DARTS loses coverage quickly, our algorithm maintains coverage that is up to twice as high. Importantly, DARTS does not capture the whole risk-coverage trade-off. Specifically, at the extreme point in the trade-off when it aims to maximize specificity, it still falls short of providing full coverage, and its predictions do not reduce to a flat classifier's predictions, i.e. it may still predict internal nodes.

Table 2: Results (mean scores) comparing the hierarchical selective threshold algorithm (Algorithm 2) with DARTS, repeated 1000 times for each model and target accuracy with a randomly drawn calibration set of 5,000 samples, applied to 1,115 models trained on ImageNet1k (meaning we evaluated each algorithm 1,115,000 times). $1 - \delta$ is set at 0.9.

| Target Accuracy (%) | Target Accuracy Error (%) | | Coverage | |
|---|---|---|---|---|
| | DARTS | Ours | DARTS | Ours |
| 70 | 14.52$\pm$1.3e-03 | **10.79$\pm$3.1e-03** | 0.97$\pm$1.4e-05 | **1.00$\pm$5.0e-05** |
| 80 | 4.86$\pm$5.2e-03 | **2.19$\pm$7.0e-03** | 0.96$\pm$5.8e-05 | **0.98$\pm$1.0e-04** |
| 90 | 0.73$\pm$9.6e-03 | **0.02$\pm$4.0e-04** | **0.88$\pm$9.9e-04** | 0.87$\pm$2.9e-05 |
| 95 | 0.69$\pm$1.4e-02 | **0.02$\pm$2.2e-04** | 0.74$\pm$2.6e-03 | **0.76$\pm$7.4e-05** |
| 99 | 0.63$\pm$4.2e-03 | **0.02$\pm$1.1e-04** | 0.22$\pm$2.0e-03 | **0.40$\pm$3.2e-04** |
| 99.5 | 0.40$\pm$1.9e-03 | **0.02$\pm$7.5e-05** | 0.13$\pm$9.8e-04 | **0.26$\pm$2.5e-04** |

Our algorithm offers additional flexibility to the user by allowing the tuning of the confidence interval $(1 - \delta)$, while DARTS does not offer such control. Figure 3 illustrates the superiority of our technique over DARTS in detail. Appendix H compares the results of an additional baseline, conformal risk control [3]. The results show that the conformal risk control algorithm exhibits a significantly higher mean accuracy error than both our algorithm and DARTS.

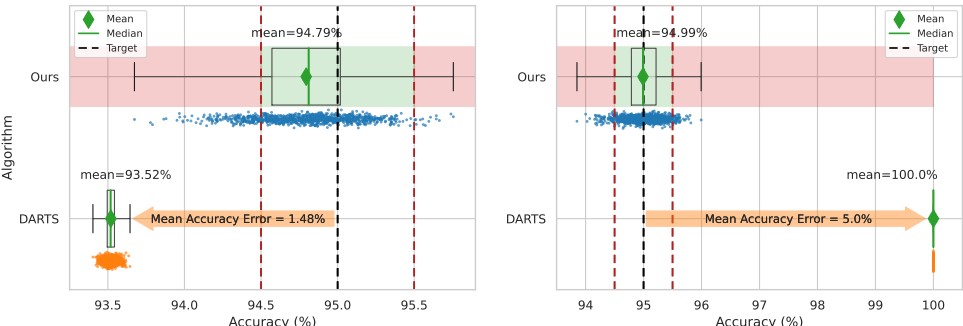

Figure 3: Individual model examples comparing the hierarchical selective threshold algorithm against DARTS, with each algorithm repeated 1000 times. The mean and median results are shown in dark green. The light green area shows the $\epsilon$ interval around the target accuracy, and the remaining area is marked in red (i.e., each repetition has a $1 - \delta$ probability of being in the green area and a $\delta$ probability of being in the red area). The target accuracy is 95% and $1 - \delta = 0.9$. In both examples, the target accuracy error of DARTS is high, and the entirety of its accuracy distribution lies outside of the confidence interval. **Left:** EVA-Giant/14 [14]. DARTS fails to meet the constraint, whereas our algorithm's mean accuracy is very close to the target. **Right:** ResNet-152 [50]. While our algorithm has a near-perfect mean accuracy, DARTS rejects all samples, resulting in zero coverage.

**Empirical Study of Training Regimes:** Inspired by [18], which demonstrated that training methods such as knowledge distillation significantly impact selective performance, we aimed to investigate whether training regimes could also contribute to hierarchical selective performance. The goal of this experimental section is to provide practitioners with valuable insights for selecting effective training regimes or pretrained models for HSC.

We evaluated the hAURC of models trained with several training regimes: (a) **Knowledge Distillation (KD)** [42, 1, 37]; (b) **Pretraining**: models pretrained either on ImageNet21k [41, 43, 56, 29, 37] or on ImageNet12k, a 11,821 class subset of the full ImageNet21k [49]; (c) **Contrastive Language-Image pretraining (CLIP)** [35]: CLIP models equipped with linear-probes, pretrained on WIT-400M image-text pairs by OpenAI, as well as models pretrained with OpenCLIP on LAION-2B [39, 8], fine-tuned either on ImageNet1k or on ImageNet12k and then ImageNet1k. Note that *zero-shot* CLIP models (i.e., CLIP models without linear-probes) were not included in this evaluation, and are discussed later in this section. (d) **Adversarial Training** [53, 44]; (e) various forms of **Weakly-Supervised** [30] or **Semi-Supervised Learning** [55, 54]. To ensure a fair comparison, we only compare pairs of models that share identical architectures, except for the method being assessed (e.g., a model trained with KD is compared to its vanilla counterpart without KD). Sample sizes vary according to the number of available models for each method. The hAURC results of all models were obtained by using the Climbing inference rule.

(1) **CLIP exceptionally improves hierarchical selective performance, compared to other training regimes**. Out of the methods mentioned above, CLIP (orange box), when equipped with a "linear-probe", improves hierarchical performance the most by a large margin. As seen in Figure 4, the improvement is measured by the relative improvement in hAURC between the vanilla version of the model and the model itself. CLIP achieves an exceptional improvement surpassing 40%. Further, its median improvement is almost double the next best methods, pretraining (purple box) and semi-supervised learning (light blue box). One possible explanation for this improvement is that the rich representations learned by CLIP lead to improved hierarchical understanding. The image-text alignment of CLIP can express semantic concepts that may not be present when learning exclusively from images. Alternatively, it could be the result of the vast amount of pertaining data.

(2) **Pretraining on larger ImageNet datasets benefits hierarchical selective performance**, with certain models achieving up to a 40% improvement. However, the improvement rates vary significantly: not all models experience the same benefit from pretraining. Semi-supervised learning also

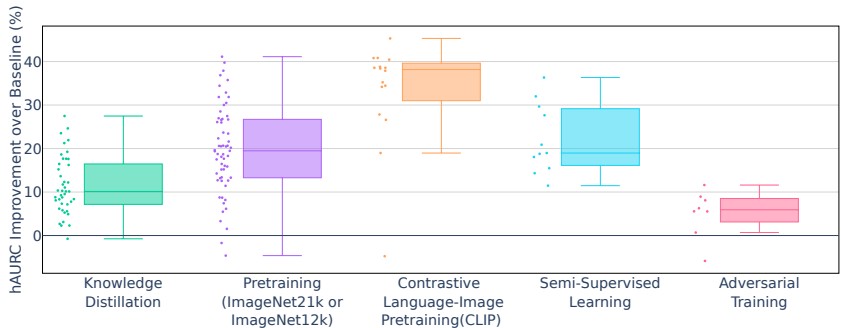

Figure 4: Comparison of different methods by their improvement in hAURC, relative to the same model's performance without the method. The number of models evaluated for each method: knowledge distillation: 42, pretraining: 61, CLIP: 16, semi-supervised learning: 11, adversarial training: 8.

shows a noticeable improvement in hierarchical selective performance.

(3) **Knowledge Distillation achieves a notable improvement**, with a median improvement of around 10%. Although not as substantial as the dramatic increase seen with CLIP, it still offers a solid improvement. This observation aligns with [18], who found that knowledge distillation improves selective prediction performance as well as ranking and calibration.

(4) **Linear-probe CLIP significantly outperforms zero-shot CLIP in HSC**: We compared pairs of models with identical backbones, where one is the original zero-shot model, and the other was equipped with a linear-probe, that is, it uses the same frozen feature extractor but has an added head trained to classify ImageNet1k. The zero-shot models evaluated are the publicly available models released by OpenAI. The mean relative improvement from zero-shot CLIP to linear-probe CLIP is 45%, with improvement rates ranging from 32% to 53%. We hypothesize that the hierarchical selective performance boost may be related to better calibration or ranking abilities. Specifically, all CLIP models with linear-probes showed significantly higher AUROC than their zero-shot counterparts, indicating superior ranking. Figure 7 in Appendix I shows these results in more detail.

Since hAURC is closely related to accuracy, we sought to determine whether its improvement could be attributed solely to gains in predictive performance from the training regimes. Our analysis, detailed in Appendix J, suggests this is not the case. Additionally, we explored a training regime specifically aimed at enhancing HSC performance. Our approach is described in Appendix K.

**HSC Improves Confidence Calibration:** We plotted the Calibration-Coverage (CC) Curves (defined in Section 2 for 1,115 ImageNet models and compared the curves produced by the Selective inference rule to those from Climbing. Figure 5 presents the aggregated CC curve for all 1,115 models across the two inference rules. The results indicate that HSC not only improves risk, but also improves calibration, with the Climbing inference rule nearly always outperforming Selective.

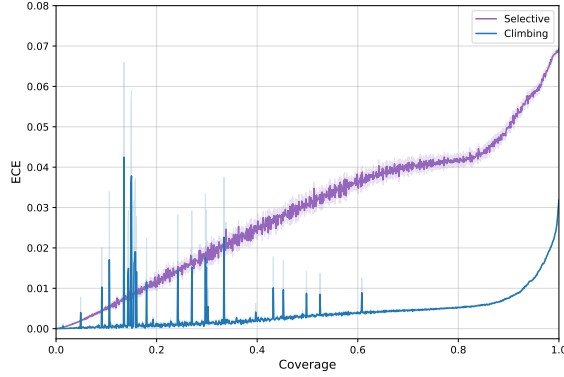

Figure 5: Aggregated (mean and SEM) CC curves of 1,115 ImageNet models.

# 6   Related Work

In selective classification, several alternative methods for confidence thresholding have been developed [5, 31, 38]. However, these methods generally necessitate some form of training. This work is focused on post-hoc methods that are compatible with any pretrained classifier, which is particularly advantageous for practitioners. Furthermore, despite the availability of other options, threshold-based rejection even with the popular Softmax confidence score continues to be widely used [18, 21], and can be readily enhanced with temperature scaling or other techniques [6, 15, 18].

Various works leverage a hierarchy of classes to improve leaf predictions of flat classifiers [51, 4, 26] but comparatively fewer works explore hierarchical classifiers. [51] optimized a "win" metric comprising a weighted combination of likelihoods on the path from the root to the leaf. [4] focused on reducing leaf mistake severity, measured by graph distance. They introduced a hierarchical cross-entropy loss, as well as a soft label loss that generalizes label smoothing. [26] claimed that both methods in [4] result in poorly calibrated models, and proposed an alternative inference method. [12] proposed the Dual Accuracy Reward Trade-off Search (DARTS) algorithm, which attempted to obtain the most specific classifier for a user-specified accuracy constraint. They used information gain and hierarchical accuracy as two competing factors, which are integrated into a generalized Lagrange function to effectively obtain multi-granularity decisions. Additional approaches include the Level Selector network, trained by self-supervision to predict the appropriate level in the hierarchy [25]. [46] proposed a loss based on [52] and performed inference using a threshold-based inference rule. Other works allow non-mandatory leaf node prediction, although not directly addressing the accuracy-specificity trade-off [36, 52]. The evaluation of hierarchical classifiers has received relatively little attention previous to our research. The performance profile of a classifier could be inferred from either the average value of a metric across the operating range or by observing operating curves that compare correctness and exactness or information precision and recall [46], or measuring information gain [12, 52]. Set-valued prediction [22] and hierarchical multi-label classification [45] tackle a similar problem to hierarchical classification by allowing a classifier to predict a set of classes. Although both approaches handle uncertainty by predicting a set of classes, the HSC framework hard-codes the sets as nodes in the hierarchy. This way, HSC injects additional world knowledge contained in the hierarchical relations between the classes, yielding a more interpretable prediction. Conformal risk control [3] is of particular interest because it constrains the prediction sets to be hierarchical. Appendix H shows a comparison of this baseline to our method.

Calibration is another important aspect of uncertainty estimation [27, 58, 18, 16]. Fisch et al. [16] demonstrate that selectively abstaining from uncertain predictions may improve calibration.

# 7   Concluding Remarks

This paper presents HSC, an extension of selective classification to a hierarchical setting, allowing models to reduce the information in predictions by retreating to internal nodes in the hierarchy when faced with uncertainty. The key contributions of this work include the formalization of hierarchical risk and coverage, hierarchical calibration, the introduction of hierarchical risk-coverage curves and hierarchical calibration-coverage curves, the development of hierarchical selective inference rules, and an efficient algorithm to find the optimal confidence threshold for a given target accuracy. Extensive empirical studies on over a thousand ImageNet classifiers demonstrate the advantages of the proposed algorithms over existing baselines and reveal new findings on training methods that boost hierarchical selective performance. However, there are a few aspects of this work that present opportunities for further investigation and improvement: (1) Our approach utilizes softmax-based confidence scores. exploring alternative confidence functions and their impact on hierarchical selective classification could provide further insights; (2) While we have identified certain training methods that boost hierarchical selective performance, the training regimes were not optimized for hierarchical selective classification. Future research could focus on optimizing selective hierarchical performance. (3) Although our threshold algorithm is effective, it could be beneficial to train models to guarantee specific risk or coverage constraints supplied by users, in essence constructing a hierarchical "SelectiveNet" [20].

## Acknowledgments

The research was partially supported by Israel Science Foundation, grant No 765/23.

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

## A  Hierarchical Severity Risk

The presence of a hierarchy allows us to measure the severity of incorrect predictions, instead of treating all mistakes as equal. For instance, classifying a Labrador as a Golden Retriever is a much milder mistake compared to classifying it as a snake. We use the coverage of the *Lowest Common Ancestor* (LCA) of the predicted node $\hat{v}$ and the label v to define a novel loss function that accounts for mistake severity: $l^H(\hat{v}, y) = 1 - \frac{\phi(LCA(\hat{v}, v))}{\phi(\hat{v})}$.

Building upon the above example, the LCA of Labrador Retriever and Golden Retriever is the node 'Retriever', which has significantly higher coverage relative to the prediction 'Golden Retriever', in contrast to 'Vertebrate', the LCA of Labrador Retriever and Snake, resulting in the former misclassification having lower risk compared to the latter.

We provide below the results of our experiments on ImageNet, complementing Table 1:

Table 3: Comparison of mean hAURC and hierarchical gain results for the selective, Max-Coverage (MC) and Climbing inference rules applied to 1,115 models trained on ImageNet1k, using the hierarchical severity risk.

| | ImageNet-1k (1,115 models) | | | |
|---|---|---|---|---|
| **Inference Rule** | **hAURC** ($\times$ **1000**) | **Hier. Gain** (%) | **hAURC** ($\times$ **1000**) | **Hier. Gain** (%) |
| **Selective** | 24.98±0.29 | | | |
| **MC** | 24.50±0.34 | 3.31±0.30 | | |
| **Climbing** | **24.04±0.31** | **4.71±0.22** | | |

The results align with those presented in the main body of the paper, where the risk is the the 0/1 loss. Climbing remains the inference rule with the best (lowest) hAURC and the highest hierarchical gain. In practice, the results of both losses are highly correlated.

## B  Max-Coverage Inference Rule Algorithm

---
**Algorithm 3** Max-Coverage Inference Rule [46]

---
**Input:** Classifier $f$, class hierarchy $H = (V, E)$, sample $x_i \in \mathcal{X}$, confidence threshold $\theta$.
**Output:** Predicted node $\hat{v}$.
    Obtain $f_v(x_i)\ \forall v \in V$
    $\hat{v} \leftarrow \arg\max_{v} \{\phi(v)|\ f_v(x_i) \geq \theta, v \in V\}$            ▷ Break ties by confidence rate

---

## C  Jumping Inference Rule

The Jumping inference rule (Algorithm 4) traverses the most likely nodes at each level of the hierarchy, until reaching $\theta$. Unlike Climbing, jumping is not confined to a single path from the root to a leaf. Although it performed better than a "hierarchically-ignorant" selective inference, it still underperformed by the other algorithms.

---

**Algorithm 4** Jumping Inference Rule

---

**Input:** Classifier $f$, class hierarchy $H = (V, E)$, sample $x_i \in \mathcal{X}$, confidence threshold $\theta$.
**Output:** Predicted node $\hat{v}$.
    Perform temperature scaling
    Obtain $f_v(x_i) \ \forall v \in V$
    $\hat{v} \leftarrow \arg\max\limits_{y \in L} f_y(x_i)$
    **while** $f_{\hat{v}}(x_i) < \theta$ **do**
        $\hat{v} \leftarrow \arg\max\limits_{v} \{f_v(x_i) | v \in V, \ height(v) = h\}$
        $h \leftarrow h + 1$
    **end while**

---

(*) If the leaves in $H$ are not all at the same depth, pad the hierarchy with dummy nodes

---

# D Hierarchy Traversal for Threshold Finding

We demonstrate an example algorithm for the Climbing inference rule that obtains $\theta_i$ for a single instance $(x_i, y_i)$.

---

**Algorithm 5** Hierarchy Traversal - Climbing

---

**Input:** Calibration sample $(x_i, y_i)$, classifier $f$, class hierarchy $H = (V, E)$, tightness coefficient $\epsilon$.
**Output:** $\theta_i$ - the minimal threshold that would have made the prediction correct.
  1: $\theta_i \leftarrow 0$
  2: $v_i \leftarrow \arg\max\limits_{y \in L} f_y(x_i)$
  3: **while** $v_i \notin \mathcal{A}(y_i)$ **do**
  4:     $\theta_i \leftarrow f_{v_i}(x_i) + \epsilon \cdot f_{\text{parent}(v_i)}(x_i)$
  5:     $v_i \leftarrow \text{parent}(v_i)$
  6: **end while**

---

# E Proof of Theorem 1

Recall that for each sample $(x_i, y_i)$, $\theta_i$ is the minimal threshold that would have made the prediction $f^H(x_i)$ hierarchically correct.

The following is based on [2]:
To avoid handling ties, we assume $\{\theta_i\}_i^n$ are distinct, and without loss of generality that the calibration thresholds are sorted: $0 \leq \theta_1 < ... < \theta_n \leq 1$.

To keep indexing inside the array limits: $\hat{\theta} \triangleq \begin{cases} \theta_{\lceil (n+1)(1-\alpha) \rceil}, & \alpha \geq \frac{1}{n+1} \\ 1, & otherwise \end{cases}$.

We also assume the samples $(x_1, y_1), ..., (x_{n+1}, y_{n+1})$ are exchangeable. Therefore, for any integer $k \in [1, n]$, we have:

$$P(\theta_{n+1} \leq \theta_k) = \frac{k}{n+1}.$$

That is, $\theta_{n+1}$ is equally likely to fall in anywhere between the calibration points. By setting $k = \lceil (n+1)(1-\alpha) \rceil$:

$$P(\theta_{n+1} \leq \theta_{\lceil (n+1)(1-\alpha) \rceil}) = \frac{\lceil (n+1)(1-\alpha) \rceil}{n+1} \geq 1 - \alpha.$$

In conformal prediction, the probability $P(\theta_{n+1} \leq \hat{\theta})$ is called "marginal coverage" (not to be confused with the hierarchical selective definition of coverage in section 2). The analytic form of marginal coverage given a fixed calibration set, for a sufficiently large test set, is shown by [47] to be

$$C(n, \alpha) = P(\theta_{n+1} \leq \hat{\theta} | \{(X_i, Y_i)\}_{i=1}^n) \sim Beta(n + 1 - l, l),$$

where $l = \lfloor (n+1)\alpha \rfloor$

From here, the CDF of the $Beta$ distribution paves the way to calculating the size $n$ of the calibration set needed in order to achieve coverage of $1 - \alpha \pm \epsilon$ with probability $1 - \delta$:

$$P(|C(n, \alpha) - (1 - \alpha)| \le \epsilon) \ge 1 - \delta.$$

**Remarks on theorem 1:**

1. To simplify, we made an implicit assumption that users can usually allocate a limited number of samples to the calibration set, and thus would prefer the threshold's optimality guarantee to hold without imposing additional requirements on the data. However, our algorithm can provide an even greater degree of flexibility: the user may choose to set the values of any three parameters out of: $\alpha, n, \epsilon, \delta$. With the three parameters fixed, the CDF of the Beta distribution can be used to compute the remaining parameter. For instance, a certain user may have an unlimited budget of calibration samples, while they require a specific error margin. In that case, the using the Beta distribution, the algorithm computes the calibration set size required for Theorem 1 to hold.

2. The output of the algorithm holds for a calibration set of any size $n$. Following the previous point, increasing $n$ does yield more stable results.

3. In the context of conformal prediction, the property stated in Theorem 1 is referred to as *marginal coverage*, not to be confused with the previous definition of coverage in section 2. This property is called marginal coverage, since the probability is marginal (averaged) over the randomness in the calibration and test points.

# F    Comparison of Inference Rules Without Temperature Scaling

Table 4: hAURC and gain results for each inference rule. Mean results of 1,115 pretrained ImageNet1k models.

|  | **Inference Rule** | hAURC ($\times 1000$) | Hierarchical Gain (%) |
|---|---|---|---|
| **No Temp. Scaling** | Selective | 42.27$\pm$0.46 | - |
|  | Max-Coverage [46] | 39.99$\pm$0.53 | 6.92$\pm$0.26 |
|  | Climbing (Ours) | **39.24$\pm$0.49** | **8.26$\pm$0.19** |
| **Temp. Scaling** | Selective | 41.16$\pm$0.46 | - |
|  | Max-Coverage | 37.21$\pm$0.52 | 11.09$\pm$0.47 |
|  | Climbing (Ours) | **36.51$\pm$0.47** | **12.51$\pm$0.16** |

# G  Threshold Algorithm Results on iNat21

Table 5: Comparison of hierarchical selective threshold algorithm (Algorithm 2) with DARTS, repeated 100 times for each model and target accuracy with a randomly drawn calibration set of 10,000 samples, applied to 6 models trained on iNat21. $1 - \delta$ is set at 0.9.

| Target Accuracy (%) | Target Accuracy Error (%) | | Coverage | |
|---|---|---|---|---|
| | DARTS | Ours | DARTS | Ours |
| 70 | $24.64 \pm 7.9\text{e-}04$ | $21.22 \pm 7.3\text{e-}04$ | $0.98 \pm 5.73\text{e-}06$ | $1.000 \pm 0.00\text{e+}00$ |
| 80 | $14.64 \pm 8.5\text{e-}04$ | $11.22 \pm 1.2\text{e-}03$ | $0.98 \pm 1.87\text{e-}06$ | $1.000 \pm 0.00\text{e+}00$ |
| 90 | $4.64 \pm 2.4\text{e-}04$ | $1.22 \pm 2.9\text{e-}03$ | $0.98 \pm 3.46\text{e-}06$ | $1.000 \pm 6.57\text{e-}05$ |
| 95 | $0.32 \pm 2.9\text{e-}02$ | $0.02 \pm 8.7\text{e-}03$ | $0.97 \pm 4.75\text{e-}04$ | $0.951 \pm 2.41\text{e-}04$ |
| 99 | $0.49 \pm 3.0\text{e-}02$ | $0.01 \pm 1.3\text{e-}03$ | $0.39 \pm 2.20\text{e-}02$ | $0.687 \pm 2.49\text{e-}03$ |
| 99.5 | $0.42 \pm 2.9\text{e-}02$ | $0.01 \pm 2.5\text{e-}03$ | $0.06 \pm 2.42\text{e-}02$ | $0.413 \pm 2.02\text{e-}03$ |

# H  Conformal Risk Control Results

Figure 6 shows the mean accuracy error of the threshold algorithms introduced in section 4 compared to the additional conformal risk control (CRC) algorithm [3]. The mean accuracy error of CRC is strikingly higher than the other algorithms, reaching a climax when the target accuracy is set to 95%.

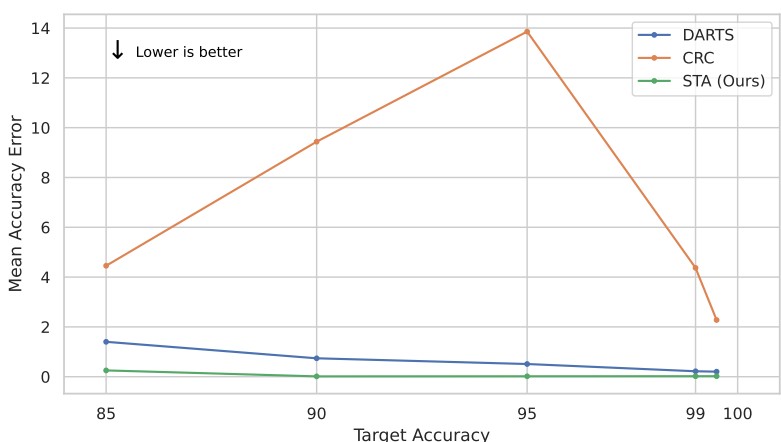

Figure 6: Mean accuracy error comparison between the hierarchical selective threshold algorithm (Algorithm 2), conformal risk control (CRC) and DARTS, repeated 1000 times for each model and target accuracy with a randomly drawn calibration set of 5,000 samples, applied to 1,115 models trained on ImageNet1k (meaning we evaluated each algorithm 1,115,000 times). $1 - \delta$ is set at 0.9.

# I  CLIP Zero Shot VS Linear Probe

Figure 7 compares the improvement in hAURC and AUROC between zero-shot CLIP models and their linear-probe counterparts. It can be easily observed that for all the tested backbones the linear-probe models have significantly better hAURC and AUROC. We encourage follow-up research to explore further the possible connection between ranking and HSC.

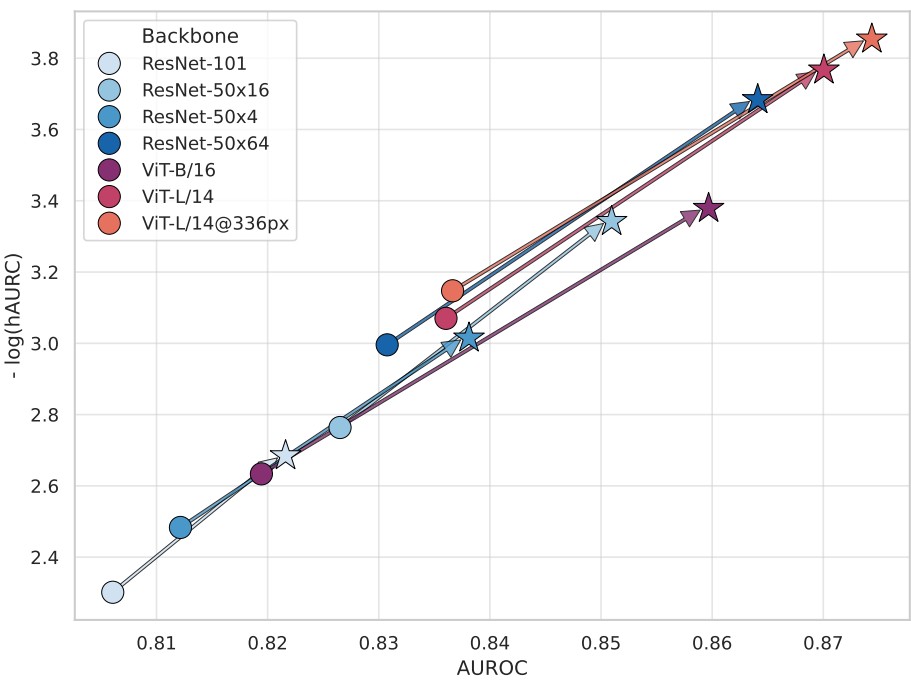

Figure 7: Comparison between zero-shot and linear-probe CLIP models sharing the same backbone. The round markers represent the zero-shot models, and the star markers represent their respective linear-probes.

## J  Training Regimes Improvement in hAURC VS Accuracy

In Table 6 we compare the relative improvement in hAURC to the relative improvement in accuracy. It can be seen that although the training regimes with the highest and lowest improvement in hAURC have the highest and lowest improvement in accuracy, the improvement in hAURC does not determine the rate of improvement in accuracy.

Table 6: Relative improvement in Accuracy compared to relative improvement in hAURC.

| Training Regime | Median Accuracy Improvement (%) | Median hAURC Improvement (%) |
|---|---|---|
| CLIP | 7.90 | 38.14 |
| Pretraining | 1.96 | 19.47 |
| SSL | 4.12 | 18.97 |
| Distillation | 1.48 | 11.73 |
| Adversarial | 0.85 | 5.92 |

## K  Training Regime to Optimize HSC Performance

Since hAURC cannot be optimized directly, we developed an alternative that could be optimized. Our best-performing method entailed training models to predict the lowest common ancestor (LCA) of pairs of samples, including identical pairs (intended for testing). To achieve this, we fine-tuned models using the hierarchical loss we proposed in Appendix A.

All models were fine-tuned on the ImageNet validation set for 20 epochs using the SGD optimizer with a warmup scheduler and a batch size of 2048 on a single NVIDIA A40 GPU. After fine-tuning models of various architectures (including ResNet50, ViTs, EfficientNet, and others) with this loss function and utilizing various other configurations, such as leveraging in-batch negatives for a richer training signal and multiple classification heads to prevent deterioration in the model's accuracy, the

improvement in hAURC we observed ranged from 3% to 5% compared to the pretrained baseline. We encourage future work to explore this direction further.

## L  Technical Details

Most experiments consider image classification on the ImageNet1k [11] validation set, containing 50 examples each for 1,000 classes. Additional evaluations were performed on the iNat21-Mini dataset [24], containing 50 examples each for 10,000 biological species in a seven-level taxonomy. For experiments requiring calibration set, it was sampled randomly from the validation set.

The evaluation was performed on 1,115 vision models pretrained on ImageNet1k [11], and 6 models pretrained on iNat-21 [24] (available in timm 0.9.16 [49] and torchvision 0.15.1 [32]), both available under the MIT license.

All experiments were conducted on a single machine with one Nvidia A4000 GPU. The evaluation of all experiments on a single GPU took approximately two weeks.

