# OpenReview forum: "Hierarchical Selective Classification"
_NeurIPS.cc/2024/Conference — NeurIPS 2024 poster_

### Official Review · Reviewer_9FfT · 2024-06-14

**Soundness:** 3
**Presentation:** 3
**Contribution:** 3
**Rating:** 6
**Confidence:** 3

**Summary:**

The authors propose hierarchical selective classification, a method that selects the hierarchical granularity of its prediction based on uncertainty.

**Strengths:**

* The paper is well-written, and the proposed method is quite intuitive.
* I like the idea that if uncertain, it makes sense to predict at a higher level of granularity.
* The theoretical results & statements are sound.
* Extensive experimental results showing the superiority of the proposed method to DARTS and a non-hierarchical baseline.
* Applicability to pre-trained models

**Weaknesses:**

* My biggest uncertainty is the similarity of this work to conformal prediction. To me, it seems that this method is very similar to conformal prediction, where the set of possible prediction sets is restricted via this pre-defined hierarchy. While, as far as I know, it has not been explored, it decreases the perceived novelty.
* A weakness of the setting rather than the method is that it assumes the knowledge of the underlying hierarchy. As such, the applicability is somewhat limited. The paper would benefit from a way to unsupervisedly learn this hierarchy, e.g. based on classes whose predicted probabilities are positively correlated.
* As also touched upon in the concluding remarks, the method is post-hoc rather than being optimized during training, thus, likely not performing up to the highest possible level.
* Minor: Line 158-159 is a worded badly, similar to "... thus, we do A. Unlike others that do A, we do A+B".

**Questions:**

* Could the authors please comment on the similarity to conformal prediction?
* As such, how is [1] related and/or different?

[1] Tyagi, Chhavi, and Wenge Guo. "Multi-label Classification under Uncertainty: A Tree-based Conformal Prediction Approach." Conformal and Probabilistic Prediction with Applications. PMLR, 2023.

**Limitations:**

Limitations are adequately discussed apart from the necessity of assuming knowledge of the underlying class hierarchy.

---

> ### Author Rebuttal · Authors · 2024-08-02
>
> Thank you for your feedback on our paper. We're glad you liked the paper and its underlying idea.
>
> *"My biggest uncertainty is the similarity of this work to conformal prediction. To me, it seems that this method is very similar to conformal prediction, where the set of possible prediction sets is restricted via this pre-defined hierarchy. While, as far as I know, it has not been explored, it decreases the perceived novelty."*
>
> Thank you for highlighting this reference. While we agree that this work merits discussion in our revision, we would like to emphasize a few key differences between our paper and conformal prediction in general, including [1]:
> 1. [1] addresses multi-label classification, while we address hierarchical classification. In multi-label classification, prediction sets consist of lists of nodes of varying sizes. In contrast, hierarchical classification always returns a single node from the hierarchy. For large hierarchies, such as those in ImageNet (1,000 leaves) or iNat21 (10,000 leaves), uncertainty could lead to very large prediction sets in multi-label classification, rendering them impractical and uninterpretable.
> 2. Unlike [1], which constructs its own tree, our approach is grounded in the use of a predefined, interpretable tree. This distinction is not just technical but fundamental to our methodology and performance.
> 3. Our method builds on selective classification, specifically addressing the trade-off between risk and coverage, an aspect not covered by [1].
>
> *"A weakness of the setting rather than the method is that it assumes the knowledge of the underlying hierarchy. As such, the applicability is somewhat limited. The paper would benefit from a way to unsupervisedly learn this hierarchy, e.g. based on classes whose predicted probabilities are positively correlated."*
>
> While it's true that our algorithms require a class hierarchy, there are many well-established hierarchies available, such as WordNet and taxonomic data for plants and animals. Consequently, popular datasets like ImageNet and iNaturalist have been built upon these hierarchical structures, providing ready-to-use trees. In cases where a class hierarchy is not provided with the dataset, it can be generated using LLMs [2,3,4,5]. Thanks to your remark, will ensure these references are included in our paper's revision, so users are aware that our methods are applicable to datasets without published hierarchies as well.
>
> *"As also touched upon in the concluding remarks, the method is post-hoc rather than being optimized during training, thus, likely not performing up to the highest possible level."*
>
> While we agree that incorporating a training regime to enhance HSC could be beneficial, our focus in this paper was on post-hoc methods, due to their immense importance and strength. As newer and more advanced models become available, our plug-and-play method, compatible with any pretrained classifier, enables state-of-the-art performance without requiring any retraining, reducing the cost of enjoying the benefits of HSC to almost zero. This feature is highly appealing to practitioners, particularly given the high costs and complexity of training modern neural networks, and even more so with recent large multimodal models. Furthermore, it is worth noting that post hoc methods are well-established and widely utilized in practice. There exists extensive literature discussing post-hoc methods, temperature scaling [6] and split conformal prediction [7] are two prime examples, among many others, such as [8,9].
>
> With that said, we agree that training models can be beneficial. In fact, we have conducted extensive experiments on model training. Since hAURC cannot be optimized directly, we developed an optimizable alternative. Our best-performing method entailed training models to predict the lowest common ancestor (LCA) of pairs of samples, including identical pairs (intended for testing). To achieve this, we developed a hierarchical loss function that penalizes the model based on the hierarchical distance between the ground truth LCA and the predicted node.
> For example: if the model is presented with an image of a Labrador and an image of a Golden Retriever, it should predict "dog" as the LCA. If the model instead predicts "animal", which is correct but less specific than "dog", it incurs a higher loss. We believed that this hierarchical loss function can improve the model's hierarchical understanding and, consequently, enhance HSC performance.
>
> After training models of various architectures with this loss and utilizing various other configurations, such as leveraging in-batch negatives for a richer training signal and multiple classification heads to prevent deterioration in the model's accuracy, the improvement in hAURC we observed ranged from 3\% to 5\% compared to the pretrained baseline. While we consider these results solid, we felt this has gone too far away from the main scope of the already packed paper. Do you think it should be included in the revision?
>
> [1] Tyagi et al. Multi-label Classification under Uncertainty: A Tree-based Conformal Prediction Approach. Conformal and Probabilistic Prediction with Applications. PMLR, 2023.
>
> [2] Chen et al. Constructing Taxonomies from Pretrained Language Models, NAACL 2021.
>
> [3] Funk et al. Towards Ontology Construction with Language Models.
>
> [4] Zeng et al. Chain-of-Layer: Iteratively Prompting Large Language Models for Taxonomy Induction from Limited Examples.
>
> [5] Li et al. Eliciting Topic Hierarchies from Large Language Models.
>
> [6] Guo et al. On Calibration of Modern Neural Networks.
>
> [7] Vladimir Vovk. Conditional validity of inductive conformal predictors.
>
> [8] Cattelan and Silva. How to Fix a Broken Confidence Estimator: Evaluating Post-hoc Methods for Selective Classification with Deep Neural Networks.
>
> [9] Galil et al. What Can We Learn From The Selective Prediction And Uncertainty Estimation Performance Of 523 Imagenet Classifiers, ICLR 2023.

---

> > ### Comment · Reviewer_9FfT · 2024-08-08
> >
> > Thank you for your response. The responses have cleared up my uncertainties, and I have adjusted my score. Regarding the end-to-end training, I believe the extension is interesting and might benefit future work that is derived from the current manuscript. As such, I would appreciate this extension in either the Appendix or as a separate work, whereby I leave it up to you to decide which you believe to be more feasible.

---

### Official Review · Reviewer_XLCN · 2024-06-19

**Soundness:** 3
**Presentation:** 3
**Contribution:** 2
**Rating:** 5
**Confidence:** 4

**Summary:**

The paper introduces a hierarchical selective classification technique that incorporates hierarchical risk and coverage. The authors additionally proposed an algorithm that guarantees target accuracy. Experimental results demonstrate the method's effectiveness.

**Strengths:**

Hierarchical selective classification is a new area and therefore the current method is one of the first techniques to deal with such problem. Its application to critical settings can be substantial.

**Weaknesses:**

•	The need of a prior tree among classes can limit its usage for complex scenarios. The construction of such tree can be a non-trivial step for the applicability of the approach.

•	The main contribution looks an extension of previous methods for the hierarchical case.

**Questions:**

•	Regarding results like in Table 2, would be possible to calibrate the coverage (as done on selective networks) for fair comparison?

•	Have the authors thought about the building the hierarchical tree structure as a pre-processing step? I asked that because such prior is key for wide applicability.

•	A well know problem of selective approaches, exposed on [1] is that given the non-differentiability of selection mechanism the binary function g is replaced by a relaxed function g: X → [0, 1], that way not performing selection during training, but instead assigning a soft instance weight to each training sample. The same effect is observed in the proposed method?

[1] Gumbel-Softmax Selective Networks, https://arxiv.org/pdf/2211.10564.

**Limitations:**

•	The requirement for a hierarchical class tree prior to using the method is a limitation.

•	The current experimental section is limited in terms of baseline approaches and datasets, making it challenging to gain a comprehensive understanding from the existing experiments.

---

> ### Author Rebuttal · Authors · 2024-08-02
>
> Thank you for your detailed feedback on our paper.
>
> *"The need of a prior tree among classes can limit its usage for complex scenarios. The construction of such tree can be a non-trivial step for the applicability of the approach."*
>
> While it's true that our algorithms require a tree structure,  there are many well-established predefined hierarchies available, such as WordNet and taxonomic data for plants and animals. Consequently, popular datasets like ImageNet and iNaturalist [1] have been built upon these hierarchical structures, providing ready-to-use trees. In cases where a hierarchical tree structure is not provided with the dataset, it can be generated using LLMs [2,3,4,5]. Thanks to your remark, will ensure these references are included in our paper's revision, so users are aware that our methods are applicable to datasets without published hierarchies as well.
>
> *"The main contribution looks an extension of previous methods for the hierarchical case."*
>
> We argue that our approach is not merely a simple extension, but a better alternative with potentially much better performance than classic selective classification. For instance, [6] boasts classifiers that achieve 99\% accuracy on ImageNet with up to 47\% coverage. However, these classifiers outright reject at least 53\% of the samples, failing to provide any useful information to the user about them.
> In contrast, our method transforms most of these otherwise rejected samples into valuable information.
> As the example in our introduction illustrates, this information could, in some scenarios, be a matter of life and death. Thus, we feel this is no mere extension to the hierarchical case, but rather, a utilization of previously unused hierarchical knowledge to turn a large number of rejections into very usable information.
>
> *"Regarding results like in Table 2, would be possible to calibrate the coverage (as done on selective networks) for fair comparison?"*
>
> Optimizing selective classification under accuracy constraints involves approaches and algorithms different than those used for optimizing under coverage constraints. There is a well known distinction in the literature between selective algorithms optimizing for an accuracy constraint [7,8] and selective algorithms optimizing for a coverage constraint [9,10]. While it is definitely possible to construct an HSC algorithm optimized for coverage, our focus in this work is on accuracy constraints. Consequently, our comparisons are made against baselines that also target accuracy. While selective networks like SelectiveNet are notable baselines for coverage calibration, they are not directly comparable to our approach due to our focus on accuracy constraints.
>
> *"A well know problem of selective approaches, exposed on [1] is that given the non-differentiability of selection mechanism the binary function g is replaced by a relaxed function g: X → [0, 1], that way not performing selection during training, but instead assigning a soft instance weight to each training sample. The same effect is observed in the proposed method?"*
>
> Thank you for bringing this paper to our attention. This seems like a more potent alternative to SelectiveNets.  We will make sure to reference it in our revision.
>
> *"The current experimental section is limited in terms of baseline approaches and datasets, making it challenging to gain a comprehensive understanding from the existing experiments."*
>
> Our experiments were conducted on two large-scale, widely accepted datasets. We considered three algorithmic baselines, and compared *each* baseline with our algorithms on over 1,100 deep neural networks. Often, we repeated the evaluations per model 1,000 times, meaning a single comparison often consisted of over a million individual comparisons. Experiments of this magnitude require a considerable amount of resources. Existing works in the domain of hierarchical classification typically consider no more than two datasets, and evaluate only few models at most. Therefore, we hope you'll agree that our experimental section is adequately comprehensive, and that the results presented clearly highlight the benefits of our methods.
>
> [1] Grant Van Horn et al. Benchmarking Representation Learning for Natural World Image Collections. https://arxiv.org/abs/2103.16483.
>
> [2] Catherine Chen, Kevin Lin, Dan Klein. Constructing Taxonomies from Pretrained Language Models, NAACL 2021. https://arxiv.org/pdf/2010.12813
>
> [3] Maurice Funk, Simon Hosemann, Jean Christoph Jung, Carsten Lutz. Towards Ontology Construction with Language Models. https://arxiv.org/pdf/2309.09898
>
> [4] Qingkai Zeng, Yuyang Bai, Zhaoxuan Tan, Shangbin Feng, Zhenwen Liang, Zhihan Zhang, Meng Jiang. Chain-of-Layer: Iteratively Prompting Large Language Models for Taxonomy Induction from Limited Examples. https://arxiv.org/pdf/2402.07386
>
> [5] Grace Li, Tao Long, Lydia B. Chilton. Eliciting Topic Hierarchies from Large Language Models. https://arxiv.org/pdf/2310.19275
>
> [6] Ido Galil, Mohammed Dabbah, Ran El-Yaniv. What Can We Learn From The Selective Prediction And Uncertainty Estimation Performance Of 523 Imagenet Classifiers, ICLR 2023. https://arxiv.org/pdf/2302.11874
>
> [7] Yonatan Geifman, Ran El-Yaniv. Selective Classification for Deep Neural Networks, NeurIPS 2017. https://papers.nips.cc/paper_files/paper/2017/hash/4a8423d5e91fda00bb7e46540e2b0cf1-Abstract.html
>
> [8] Jia Deng; Jonathan Krause; Alexander C. Berg; Li Fei-Fei. Hedging your bets: Optimizing accuracy-specificity trade-offs in large scale visual recognition.
>
> [9] Yonatan Geifman, Ran El-Yaniv. SelectiveNet: A Deep Neural Network with an Integrated Reject Option, ICML 2019. https://proceedings.mlr.press/v97/geifman19a/geifman19a.pdf
>
> [10] Guy Bar-Shalom, Yonatan Geifman, Ran El-Yaniv. Window-Based Distribution Shift Detection for Deep Neural Networks, NeurIPS 2023. https://proceedings.neurips.cc/paper_files/paper/2023/hash/4791edcba96fbd82a8962b0f790b52c9-Abstract-Conference.html

---

> > ### Comment · Reviewer_XLCN · 2024-08-11
> >
> > I appreciate the authors' rebuttal. I would raise my score from 4 to 5 in response to the authors' rebuttal. I will also discuss with the other fellow reviewers and/or AC.

---

### Official Review · Reviewer_vmdY · 2024-07-02

**Soundness:** 3
**Presentation:** 2
**Contribution:** 3
**Rating:** 6
**Confidence:** 3

**Summary:**

The paper introduces a new framework for selective classification called hierarchical selective classification. In a setting where a hierarchy in the classification task is present, the authors devise a selection strategy that considers confidence at different levels of the classification hierarchy. Extensive experimental analysis is performed over 1115 ImageNet classifiers.

**Strengths:**

The main strengths of the paper are:

1. the idea of applying selective classification in a hierarchical setting is novel;
2. the theoretical analysis relies on conformal prediction, which guarantees the soundness of the results;
3. the proposed framework can impact high-risk settings, as shown in the healthcare example.

**Weaknesses:**

Overall, I think the paper is solid. My main concern is that the empirical evaluation could be improved, especially regarding motivations and attention to detail.
A few examples:
* I do not fully understand why the authors focus so much on showing how different training regimes affect HSC performance. I guess this improves the overall predictive performance of the (hierarchical) classifier, which is expected to impact the HSC task positively.
* As the authors correctly claim, the training regimes were not optimized for hierarchical selective classification. Despite the clear computation-wise motivation, I argue that including regimes optimized for HSC would make the empirical evaluation sounder.
* a few lines are off: for instance, I would argue that line 279, i.e.,
>CLIP achieves an exceptional improvement, surpassing 40%
>
 does not match what is shown in Figure 4 (which shows an improvement below 40%).

**Questions:**

I have a few questions/remarks regarding the paper.

* Q1. I think the authors are not discussing an implicit (and, in my opinion, quite relevant) assumption of their strategy, i.e. the errors at different levels of the hierarchy are assumed to be the same. However, I argue this is not exactly the case in real life. For example, failing to distinguish a golden retriever from a labrador differs from failing to distinguish a dog from a spider. Can the authors elaborate on this point?
* Q2. Can the authors discuss the points I highlighted as the main weakness?

**Limitations:**

The paper briefly discusses limitations. I think this section could be expanded, e.g. considering Q1.

---

> ### Author Rebuttal · Authors · 2024-08-02
>
> Thank you for your constructive feedback.
>
> *"I do not fully understand why the authors focus so much on showing how different training regimes affect HSC performance. I guess this improves the overall predictive performance of the (hierarchical) classifier, which is expected to impact the HSC task positively."*
>
> The methods presented in this paper are compatible with any pre-trained base classifier. As such, we were particularly interested in exploring how to guide practitioners towards choosing base classifiers with high HSC performance.
> The experiments in this section draw inspiration from [1], who found that training regimes such as knowledge distillation significantly impact selective performance. Their findings provide valuable guidance for practitioners, helping them choose models trained using regimes that result in high selective performance. Following their findings, we set out to find if training regimes could contribute to hierarchical selective performance as well, and if our findings align with those presented in [1].
>
> We initially shared your concern that the improvement in hAURC could be attributed to overall improvement in predictive performance resulting from the training regimes. This led us to analyze the improvement in accuracy compared to the improvement in hAURC. From the results below, it seems to us that the improvement in hierarchical performance is significantly more pronounced than the one in accuracy. Furthermore, We see examples of regimes with similar improvements in accuracy having significant differences in hierarchical performance improvements (e.g., "Pretraining" and "Distillation").
>
> We present in the table below the mean improvement in accuracy compared to hAURC.
>
> | Training Regime | Median Accuracy Improvement (\%) | Median hAURC Improvement (\%) |
> |-----------------|---------------------------------|------------------------------|
> | CLIP            | 7.90                            | 38.14                        |
> | Pretraining     | 1.96                            | 19.47                        |
> | SSL             | 4.12                            | 18.97                        |
> | Distillation    | 1.48                            | 11.73                        |
> | Adversarial     | 0.85                            | 5.92                         |
>
> We initially chose not to include this analysis in the paper due to concerns about the paper being already packed with too much information. Do you think it would be preferable to include these results in the revision?
>
> *"As the authors correctly claim, the training regimes were not optimized for hierarchical selective classification. Despite the clear computation-wise motivation, I argue that including regimes optimized for HSC would make the empirical evaluation sounder."*
>
> While we agree that incorporating a training regime to enhance HSC could be beneficial, our focus in this paper was on post-hoc methods, due to their immense importance and strength.
> Our method’s strength lies in its compatibility with any pretrained classifier. As newer and more advanced models become available, our plug-and-play method enables access to state-of-the-art classifiers without requiring any retraining, reducing the cost of enjoying the benefits of HSC to almost zero. This feature is highly appealing to practitioners, particularly given the high costs and complexity of training modern neural networks, and even more so with recent large multimodal models.
> Furthermore, it is worth noting that post hoc methods are well-established and widely utilized in practice. Thanks to their effectiveness, there exists extensive literature discussing post-hoc methods, temperature scaling [2] and split conformal prediction [3] are two prime examples of post-hoc practical approaches, among many others, such as [1,4].
>
> With that said, we agree that training models can be beneficial for improving HSC performance. In fact, we have conducted extensive experiments on model training. Since hAURC cannot be optimized directly, we developed an alternative that could be optimized. Our best-performing method entailed training models to predict the lowest common ancestor (LCA) of pairs of samples, including identical pairs (intended for testing). To achieve this, we developed a hierarchical loss function that penalizes the model based on the hierarchical distance between the ground truth LCA and the predicted node.
> For example: if the model is presented with an image of a Labrador and an image of a Golden Retriever, it should predict "dog" as the LCA. If the model instead predicts "animal", which is correct but less specific than "dog", it incurs a higher loss. We believed that this hierarchical loss function can improve the model's hierarchical understanding and, consequently, enhance HSC performance.
>
> After training models of various architectures (including ResNet50, ViTs, EfficientNet, and others) with this loss function and utilizing various other configurations, such as leveraging in-batch negatives for a richer training signal and multiple classification heads to prevent deterioration in the model's accuracy, the improvement in hAURC we observed ranged from 3\% to 5\% compared to the pretrained baseline. While we consider these results solid, we felt this has gone too far away from the main scope of the already packed paper and decided to leave it outside of it. We would value your feedback, do you think this method and results should be included in the revision?
>
> *"I would argue that line 279 does not match what is shown in Figure 4 (which shows an improvement below 40\%)"*
>
> Thank you for pointing this out, we meant to say some instances of CLIP surpass 40\%. We will report the median of the sample (38\%) instead.

---

> > ### Comment · Reviewer_vmdY · 2024-08-10
> >
> > I thank the authors for their responses.
> > I have a couple of remarks/comments:
> >
> > > From the results below, it seems to us that the improvement in hierarchical performance is significantly more pronounced than the one in accuracy.
> >
> > If I get how improvements are computed (i.e. $\Delta(Metric)=\frac{|Metric_{new} - Metric_{orig}|}{Metric_{orig}}$), I think it is not that straightforward to compare the % improvements of two different metrics. I feel this is affected by the original baseline value starting point. Given an absolute improvement, the relative improvement increases as the baseline value decreases, and vice versa. I.e., an absolute improvement of $\delta$ on baseline values $b_1, b_2, b_1 << b2$  results in relative improvements $r_1, r_2, r_1 >> r_2$.
> >
> > Since the lower the hAURC, the better, and the higher the accuracy, the better (as in the just presented example); seeing higher percentage improvements might be misleading, and I would be cautious about the conclusions here.
> >
> > > We initially chose not to include this analysis in the paper due to concerns about the paper being already packed with too much information. Do you think it would be preferable to include these results in the revision?
> >
> > I think these results could be safely added to the Appendix, as they could explain why we see improvements in hAURC through different training regimes.
> >
> > > However, in practice, the differences are smaller than we anticipated and are very highly correlated (which is interesting by itself). Do you think we should include these results and others with this risk in the revision?
> >
> > Concerning the results w.r.t. the hierarchical loss, the authors could add the results in the Appendix.
> >
> > In light of these comments, I am still in favour of a positive score for this paper.

---

> ### Author Response · Authors · 2024-08-07
> **Rebuttal Part 2 for Reviewer vmdY**
>
> *"Q1. I think the authors are not discussing an implicit (and, in my opinion, quite relevant) assumption of their strategy, i.e. the errors at different levels of the hierarchy are assumed to be the same. However, I argue this is not exactly the case in real life. For example, failing to distinguish a golden retriever from a labrador differs from failing to distinguish a dog from a spider. Can the authors elaborate on this point?"*
>
> That's a great question! In fact, we conducted all of our experiments using a hierarchical risk that considers scenarios like the ones you've illustrated. We chose not to include these results in the paper due to space constraints and for the sake of simplicity and readability.
>
> We developed a hierarchical risk that considers mistake severity with respect to the hierarchy, as follows:
>
> $R_h = 1 - \frac{\phi(LCA(\hat{v}, v))}{\phi(\hat{v})}$
>
> Where $v$ is the ground truth node, $\hat{v}$ is the predicted node, $\phi(x)$ is the coverage of node $x$, and $LCA(x,y)$ is the lowest common ancestor of nodes $x$ and $y$. The most severe mistake occurs when $LCA(v,\hat{v})$ is the root of the hierarchy, resulting in a risk of 1. The higher the coverage of $LCA(\hat{v},v)$, the more $\hat{v}$ has in common with $v$ in hierarchical terms.
> One can safely assume that in any plausible hierarchy the LCA of (labrador, golden retriever), for example, dog, has significantly higher coverage compared to the LCA of (dog, spider), resulting in the former misclassification having lower risk compared to the latter.
>
> We provide below some of the results of our experiments on ImageNet, complementing Table 1 in the paper (all other tables and figures can also be made with this type of loss):
>
> |                | ImageNet-1k (1115 models) |               |
> |:--------------:|:-------------------------:|:-------------:|
> | Inference Rule |           hAURC           |   Hier. Gain  |
> |                |       $\times$ 1000       |      (\%)     |
> | Selective      | 24.98$\pm$0.29            | -             |
> | MC             | 24.50$\pm$0.34            | 3.31$\pm$0.30 |
> | Climbing       | 24.04$\pm$0.31            | 4.71$\pm$0.22 |
>
> The results align with those presented in the paper, where the risk is 0/1 loss. Climbing remains the inference rule with the best (lowest) hAURC and the highest hierarchical gain. It's theoretically possible to create examples where the hierarchical risk diverges from the 0/1 loss, such that models making less severe mistakes are penalized less by the hierarchical risk. However, in practice, the differences are smaller than we anticipated and are very highly correlated (which is interesting by itself). Do you think we should include these results and others with this risk in the revision?
>
> [1] Ido Galil, Mohammed Dabbah, Ran El-Yaniv. What Can We Learn From The Selective Prediction And Uncertainty Estimation Performance Of 523 Imagenet Classifiers, ICLR 2023. https://arxiv.org/pdf/2302.11874
>
> [2] Guo et al. On Calibration of Modern Neural Networks.
>
> [3] Vladimir Vovk. Conditional validity of inductive conformal predictors.
>
> [4] Cattelan and Silva. How to Fix a Broken Confidence Estimator: Evaluating Post-hoc Methods for Selective Classification with Deep Neural Networks.

---

### Official Review · Reviewer_h9cA · 2024-07-10

**Soundness:** 3
**Presentation:** 3
**Contribution:** 3
**Rating:** 6
**Confidence:** 3

**Summary:**

The paper proposes an extension of selective classification following a class hierarchy to reduce the specificity of model prediction when there is a high uncertainty. In particular, if the prediction confidence of a class is smaller than a predefined threshold, the proposed algorithm would proceed towards a higher class level in the hierarchical structure (the parent node), until the confidence of the considering node exceeds that threshold. The paper also formulises hierarchical risk and coverage, so that the area under curve can be used as a metric to benchmark different selective classification methods. An extensive number of pretrained classifiers on ImageNet dataset are then used to evaluate the proposed method and show promising results. The paper also include a PAC-like theoretical result, so that when finding the optimal threshold, one can select appropriate hyper-parameters to achieve their desired outcome with certain confidence level.

**Strengths:**

The paper goes into details to provide an adequate background about selective classification, the definition of heirarchical risk and coverage as well as its area under curve as a metric to quantify the performance of hierarchical-based selective classification. It also links to previous studies in the same subfield. In general, the paper is well written and easy to follow.

The paper also includes a theoretical result on the guarantee of the learning algorithm when one wants to find the optimal thresholding value for their hierarchical selective classification. This simple theoretical results does strengthen the paper.

The paper also include an extensive number of experiments and ablation studies to provide insights into the newly-proposed method.

**Weaknesses:**

The paper relies on the setting with the following assumptions:
- It is an inference rule. This means that the algorithm is used at test time only. If this could be even integrated into training is a plus.
- It needs a validation set to find the optimal hyper-parameter $\theta$, or the threshold (partly mentioned in the conclusion). It is understandable because there is no training involve here, so there is a need for that. However, in some cases, there may not be additional data available.

**Questions:**

Could the authors clarify if it can also be integrated into training a whole model to perform hierarchical selective classification?

---

> ### Author Rebuttal · Authors · 2024-08-02
>
> Thank you for your positive reply,
>
> *"It is an inference rule. This means that the algorithm is used at test time only. If this could be even integrated into training is a plus.", "Could the authors clarify if it can also be integrated into training a whole model to perform hierarchical selective classification?"*
>
> While we agree that incorporating a training regime to enhance HSC could be beneficial, our focus in this paper was on post-hoc methods, due to their immense importance and strength. Our method’s strength lies in its compatibility with any pretrained classifier. As newer and more advanced models become available, our plug-and-play method enables access to state-of-the-art classifiers without requiring any retraining, reducing the cost of enjoying the benefits of HSC to almost zero. This feature is highly appealing to practitioners, particularly given the high costs and complexity of training modern neural networks, and even more so with recent large multimodal models.
> Furthermore, it is worth noting that post hoc methods are well-established and widely utilized in practice. Thanks to their effectiveness, there exists extensive literature discussing post-hoc methods, temperature scaling [1] and split conformal prediction [2] are two prime examples of post-hoc practical approaches, among many others, such as [3,4].
> With that said, we agree that training models can be beneficial for improving HSC performance. In fact, we have conducted extensive experiments on model training. Since hAURC cannot be optimized directly, we developed an alternative that could be optimized. Our best-performing method entailed training models to predict the lowest common ancestor (LCA) of pairs of samples, including identical pairs (intended for testing). To achieve this, we developed a hierarchical loss function that penalizes the model based on the hierarchical distance between the ground truth LCA and the predicted node.
> For example: if the model is presented with an image of a Labrador and an image of a Golden Retriever, it should predict "dog" as the LCA. If the model instead predicts "animal", which is correct but less specific than "dog", it incurs a higher loss. We believed that this hierarchical loss function can improve the model's hierarchical understanding and, consequently, enhance HSC performance.
> After training models of various architectures (including ResNet50, ViTs, EfficientNet, and others) with this loss function and utilizing various other configurations, such as leveraging in-batch negatives for a richer training signal and multiple classification heads to prevent deterioration in the model's accuracy, the improvement in hAURC we observed ranged from 3\% to 5\% compared to the pretrained baseline. While we consider these results solid, we felt this has gone too far away from the main scope of the already packed paper and decided to leave it outside of it. We would appreciate your opinion, do you think it's preferable to include this method and results in the revision?
>
> *"It needs a validation set to find the optimal hyper-parameter, or the threshold (partly mentioned in the conclusion). It is understandable because there is no training involve here, so there is a need for that. However, in some cases, there may not be additional data available."*
>
> While we acknowledge that collecting a calibration set (or reducing it from the training set) comes at a cost, we believe this limitation is "soft" in our case, since, as mentioned in the note following Theorem 1 and in the remarks in Appendix D, our guarantees hold for calibration sets of any size. Even for a small calibration set, our algorithm provides users with the flexibility to set the other parameters ($\delta$ and $\epsilon$) according to their requirements and constraints. For example: a user with a low budget for a calibration set who may be, perhaps, more interested in controlling $\delta$ can still achieve a reasonable constraint by relaxing $\epsilon$ instead of increasing the calibration set size.
> Additionally, as we mentioned above, post-hoc methods that require an additional calibration set are widely accepted and used by the community thanks to their efficacy [1,2,3,4].
>
> [1] Chuan Guo, Geoff Pleiss, Yu Sun, Kilian Q. Weinberger. On Calibration of Modern Neural Networks. https://arxiv.org/pdf/1706.04599
>
> [2] Vladimir Vovk. Conditional validity of inductive conformal predictors. https://arxiv.org/pdf/1209.2673
>
> [3] Luís Felipe P. Cattelan, Danilo Silva. How to Fix a Broken Confidence Estimator: Evaluating Post-hoc Methods for Selective Classification with Deep Neural Networks. https://arxiv.org/abs/2305.15508
>
> [4] Ido Galil, Mohammed Dabbah, Ran El-Yaniv. What Can We Learn From The Selective Prediction And Uncertainty Estimation Performance Of 523 Imagenet Classifiers, ICLR 2023. https://arxiv.org/pdf/2302.11874

---

> > ### Comment · Reviewer_h9cA · 2024-08-12
> > **Comments by Reviewers h9cA**
> >
> > Thank you, the authors, for clarifying my concerns. I have a positive view of the paper, although I am not very familiar to the research field. Hence, I keep my rating as is.

---

### Comment · Area_Chair_ne2v · 2024-08-11

Dear Reviewers,

The discussion period ends within 3 days. Authors made tremendous efforts on providing responses to your concerns. So, if you haven’t yet, please check responses and express your opinions whether you want to initiate discussion or accept the authors’ responses.


Best,
Your AC

---

### Decision · Program_Chairs · 2024-09-25

**Decision:**

Accept (poster)

**Comment:**

All reviewers appreciated this paper’s contributions. In particular, this paper proposes a novel hierarchical selective classification, which is clearly distinguished to conventional selective classification and conformal prediction. Moreover, by introducing hierarchical structures into consideration, it effectively exploits information from rejected samples by the conventional selective classification. Also the hierarchical structure provides a favorable property: monotonicity in correctness and coverage, as in conformal prediction. Thus, I also vote for acceptance.

In the final manuscript, please add promised contents discussed during the rebuttal period.